# CSGAN: a consistent structural GAN for AS-OCT image despeckling by image translation

## Abstract

Anterior segment optical coherence tomography (AS-OCT) is a recent imaging technique for visualizing the physiological structure of the anterior segment. The speckle noise inherited in ASOCT images degrades the visual quality and hampers the subsequent medical analysis. Previous work was devoted to removing the speckles and acquiring satisfying images. According to the clinical requirements, it might be desirable to maintain locally higher data fidelity instead of enforcing visually appealing but rather wrong image structural features. Catering to this expectation, we propose a Consistent Structural Generative Adversarial Network (CSGAN) to learn the clean style of low-speckle in repeated AS-OCT images and simultaneously preserve the tiny but vital structural knowledge among the latent feature, spatial and frequency domains. Specifically, we design a latent constraint into the generator to capture the inherent content in the feature domain and adopt the perceptual similarities to directly preserve structural detail in the spatial dimension. Besides, we introduce a focal frequency scheme that adaptively represents and distinguishes hard frequencies to compensate for the spatial loss and refine the generated image to improve image quality. Finally, the experimental results demonstrate that the CSGAN can achieve satisfactory despeckling results with preserving structural details on the AS-Casia dataset.

**Keywords:** AS-OCT, image despeckling, structural consistency, GAN.

## 1. Introduction

Anterior segment optical coherence tomography (AS-OCT), a non-invasive imaging technique, is widely utilized in ophthalmology to diagnose anterior segment disorders, such as glaucoma, cataract, and corneal diseases. However, AS-OCT images inevitably suffer from speckle noise, which obscures subtle morphological details and impacts the clinical diagnosis. Therefore, commercial AS-OCT scanners generally average multiple B-scans captured repeatedly at the same location to suppress the speckle noise. This method has two main limitations: first, averaging multiple B-scans requires longer scanning times, making the patients uncomfortable. Second, involuntary movements or sample motions can lead to motion artifacts or structures lost in the repeated low-speckle images.

Although the averaged images easily lost the structural information, their clean style is desirable. Therefore, the despeckling image task can be cast as a translation problem. Specifically, the image-to-image translation scheme attempts to learn the structure knowledge and context of the speckled image and incorporate the style of averaged images into the restored images. Previous work based on generative models, including pix2pix(Isola et al., 2017) can achieve image translation while easily omitting the tiny but vital structures. To meet the requirement of the clinical practice, we incorporate some strategies into the style learning network for preserving structural details among feature domains, spatial dimensions, and frequency space in AS-OCT image despeckling.

## 2. Methods

**CSGAN architecture:** Driven by the fact that the conditional adversarial networks have already taken a significant step in the image-to-image translation, CSGAN adopts the general generator ($G$) and discriminator ($D$) to achieve the pix-level representations. As shown in Figure. 1, The $G$ consists of encoder ($EC$) and decoder ($DC$) while the $D$ directly utilizes the PatchGAN for the classification of image pairs as real or fake based on image patches rather than the whole image, and it can be understood as a form of texture/style loss from speckled/repeated images. The $EC$ adopt 2 downsampling operations consisting of Convolution-Batch Normalization-ReLU layer and several residual taking the form of convolution-BatchNorm and Convolution-BatchNorm-ReLU. Then, we design a latent variable restriction as a regularization term to offset the impact of inaccurate encoding and assist the network in capturing the underlying semantic structural knowledge. Finally, we use the fractionally-srided Convolution layer to achieve the upsampling operation.

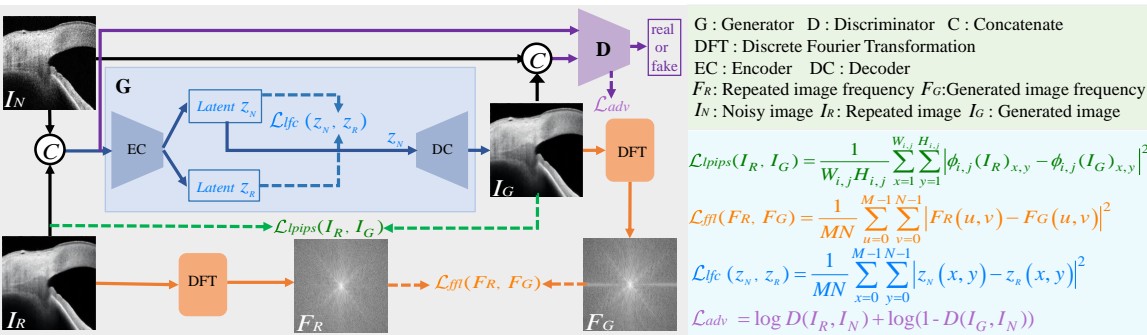

Figure 1: The framework of the proposed CSGAN.

**Objective function:** The generative network demonstrate the powerful representation ability in low-level image processing task while it always neglects some tiny but vital details for clinical practice. To balance the speckle suppression and structure consistency, we first exploit the latent feature constraint $\mathcal{L}_{lfc}$ to capture structural semantics in the latent feature maps and allow some slight variation to improve the generalization ability of network. Then, the perceptual loss $\mathcal{L}_{lpips}$ (Zhang et al., 2018) is explicitly adopted to learn perceptual image patch similarity and incorporate structural texture and minimize the structural gap the speckled and generated image in the spatial domain. Moreover, speckle noise and structural details, including edge features and texture grain, generally manifest as high-frequency sections in the frequency domain. Therefore, restoring intact and precise useful frequency components and suppressing the speckled spectrum is challenging. To solve this bottleneck, we adopt the focal frequency scheme to enforce the network locating the hard frequencies and utilize weighted spectrum value to strengthen the structural knowledge in the frequency domain dynamically. Specifically, a focal frequency loss $\mathcal{L}_{ffl}$ (Jiang et al., 2021) is adopted to adaptively focus the model on the frequency components that are hard to deal with but can be pivotal for ameliorating quality in the frequency space. The AS-OCT images are transformed into the frequency domain by 2D-DFT and represented using amplitude $u$ and phase $v$ such that each frequency value can be mapped to a Euclidean vector in a two-dimensional space. Finally, like the general GAN, the primary objective

of CSGAN also contains adversarial loss $\mathcal{L}_{adv}$ in the PatchGAN discriminators to distinguish the classification of style or structural structure. Therefore, the CSGAN can capture the despeckling style and preserve the tiny but vital structural details by characterizing structural similarity among spatial domain, latent feature maps, and frequency space.

## 3. Experimental results and analysis

We collected the AS-casia dataset from the CASIA2 ophthalmology device. The dataset provides 184 noisy and 184 clean images by averaging 16 repeated B-scans at the same position. All images are views of the anterior segment (AS) structure, including the ciliary muscle, iris, and anterior chamber angle. The visual comparison among the noisy images, repeated images, and despeckling results of the proposed CSGAN is shown in Figure. 2. We can observe that the repeated images easily get trapped in edge artifacts at the border of the iris (see the orange arrow in the enlarged green region) and loss of detail (see the blue arrow) caused by involuntary movements during the acquisition. At the same time, the despeckling result can capture the structure knowledge in noisy images and learn the style of repeated images owing to the designing of structure consistency in CSGAN and the style understanding ability in PatchGAN-discriminator.

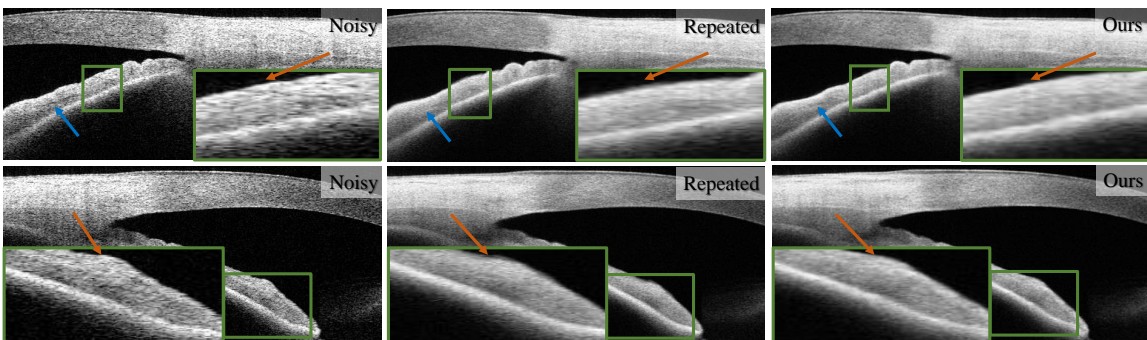

Figure 2: The visual comparison of despeckling result.

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
