# OpenReview forum: "CSGAN: a consistent structural GAN for AS-OCT image despeckling by image translation"
_MIDL.io/2023/Short_Paper_Track — MIDL 2023 Short paper track Poster_

### Official Review · Reviewer_epkh · 2023-04-24
**Review of: CSGAN: a consistent structural GAN for AS-OCT image despeckling by image translation**

**Rating:** 6
**Confidence:** 4

**Review:**

The authors propose a despeckling method using Image-to-Image GANs, intended to replace multiframe averaging in OCT.

Overall, this submission seems like a good start. We should see in later submissions comparisons to:
1) off the shelf baselines (BM3D, local smoothing, etc)
2) generic Image-to-Image methods, ablating each of the innovations of this paper (so, the fft loss, the perceptual loss, etc.)
Importantly, it would be great to have PSNR or similar for a numerical comparison.

As it stands I think it's fine for an abstract, and the architecture itself appears quite built-out, but it cannot be understood from the current experiments if this is an actual improvement or not. Thus, I think the section to focus on moving forward is the experimental section: including a validation of your method versus baselines, and validation of the particular innovations made will make the next submission of this work much stronger.

---

### Official Review · Reviewer_dM4S · 2023-04-25
**Using GANs to de-speckle OCT images while preserving relevant structure**

**Rating:** 6
**Confidence:** 4

**Review:**

The paper proposes to use GANs to denoise OCT images. To ensure clinical accuracy, the authors develop losses that preserve frequencies that correspond to anatomical structures. This is an interesting idea that might apply to other modalities as well. The key challenge is how to identify the image structure that corresponds to relevant information vs. noise/speckle.